# Properties in Langmuir Monolayers and Langmuir-Blodgett Films of a Block Copolymer Based on *N*-Isopropylacrylamide and 2,2,3,3-Tetrafluropropyl Methacrylate

**DOI:** 10.3390/polym14235193

**Published:** 2022-11-29

**Authors:** Olga Zamyshlyayeva, Zarina Shaliagina, Maria Simonova, Alexander Filippov, Maxim Baten’kin

**Affiliations:** 1Department of High Molecular Compounds and Colloidal Chemistry, Faculty of Chemistry, Lobachevsky State University, Gagarina pr. 23, Nizhny Novgorod 603950, Russia; 2Institute of Macromolecular Compounds Russian Academy of Science, Bolshoy pr. 31, Saint-Petersburg 199004, Russia; 3G.A. Razuvaev Institute of Organometallic Chemistry of Russian Academy of Sciences, 49 Tropinin Street, Nizhny Novgorod 603950, Russia

**Keywords:** block copolymer, synthesis, poly-*N*-isopropylacrylamide, Langmuir monolayer, Langmuir-Blodgett films

## Abstract

The amphiphilic block copolymer poly(*N*-isopropylacrylamide)–Ge(C_6_F_5_)_2_–poly(2,2,3,3-tetrafluoropropyl methacrylate) was prepared by the reaction of chain transfer to *bis*-(pentafluorophenyl)germane during the polymerization of *N*-isopropylacrylamide and the subsequent postpolymerization of isolated functional polymers in 2,2,3,3–tetrafluoropropyl methacrylate. The conversion of the block copolymer was 68% and the molecular weight of the sample was 490,000 g/mol. The colloidal chemical properties of Langmuir monolayers and Langmuir-Blodgett films of synthesized block copolymer have been studied. For comparison, a functional polymer, namely, poly-*N*-isopropylacrylamide with terminal –Ge(C_6_F_5_)_2_H group, was synthesized and studied. The concentrations of spreading solutions were selected and the effect of subphase acidity on the formation of monolayers of macromolecules of the block copolymer was studied. It was found that regardless of the acidity of the subphase, high pressure of fracture of films are characteristic of monolayers of collapse pressures π_max_ = (48–61) mN/m. The morphology of the Langmuir-Blodgett films of functional polymer exhibit isolated elongated micelles with high densities in the form of “octopus” on the periphery of which there are terminal hydrophobic groups. For the Langmuir-Blodgett film of block copolymer, a comb-like structure is observed with characteristic protrusions.

## 1. Introduction

Currently, poly-*N*-isopropylacrylamide (PNIPAAm) remains the most studied polymer, demonstrating a phase transition from an insoluble to a soluble state at a certain temperature. It exhibits the so-called lower critical dissolution temperature (LCST) behavior [1,2]. Below LCST, PNIPAAm is hydrophilic, and its copolymerization with hydrophobic or hydrophilic comonomers changes its properties, lowering or increasing the LCST of its aqueous solutions, respectively. At present, the copolymerization of the NIPAAm monomer makes it possible to obtain copolymers of various structures with different phase transition temperatures, which have already found applications in medicine [3,4,5].

Piskin E. et al. [6,7,8,9] described the synthesis and determined the characteristics of PNIPAAm–SCH_2_COOH samples with a terminal carboxyl group, differing in molar mass (MM) and LCST. The possibility of synthesis of block copolymers of PNIPAAm with polyethyleneimine (PEI) (*M_n_* = 2000 g/mol) was also demonstrated. Block copolymers were prepared by the reaction of the terminal carboxylic acid groups of PNIPA with PEI, using 1-ethyl-3-(3-(dimethylamino)propyl)carbodiimide (EDAC) in an aqueous solution. The viscosity MM of block copolymers, obtained using the well-known constants *K_η_* = 0.0003 and α = 0.64 in the Mark–Kuhn–Houwink equation, ranged from 23,000 to 57,000 g/mol. An increase in the MM of the PNIPAAm block led to a change in the LCST from 43.1 to 34.9 °C at pH = 4.0 and from 42.0 to 33.4 °C at pH = 7.4 [10].

Amphiphilic diblock copolymers of PNIPAAm with polystyrene (PS) and poly-tert-butyl methacrylate (PBMA), differing in the MM of the blocks, were obtained by RAFT polymerization using 4-cyanopentanoic acid dithiobenzoate as a chain transfer agent [11]. In water, copolymers with hydrophobic blocks formed micelles for a long time. It has been established that the reason of the formation of micellar structures is the dehydration of PNIPAAm chains, which is characteristic of all studied copolymers in a wide temperature range. In addition, it was found that the surface of the PS and PBMA of the micelle cores is covered with hydrophobic groups in such a way that they prevent the compression of the hydrophilic shell of the PNIPAAm core. The micelles were colloidally stable and did not precipitate from water at high temperatures.

A thermo- and pH-responsive copolymer of *N*-isopropylacrylamide with maleic acid was studied by the light scattering and turbidimetry [12,13]. Aqueous solutions with pH values from 1.8 to 10.9 and in the concentration range from 0.001 to 0.015 g/cm^3^ were investigated. At all pH values and concentrations, phase separation was observed at temperatures *T* > 33 °C. The temperatures of the onset of the phase separation and the width of this interval increased with decrease in copolymer concentration and increase in pH. Three types of spices were found in solutions, namely, macromolecular unimers, micellar structures, and loose aggregates. At heating, their size and fraction in solutions were changed.

Copolymers of PNIPAAm with fluorine-containing comonomers are of particular interest. The need to create methods for the synthesis of block and graft copolymers based on fluorine-containing compounds is primarily due to the surface properties of these materials, which are characterized by low values of surface energy [14,15]. In addition, the presence of fluorinated groups with a strong cohesive potential in the composition of block copolymers leads to an increased propensity of macromolecules for self-organization in solution [16,17,18,19,20]. Diphility of components in copolymers based on NIPAAm and fluorinated comonomers is the cause of self-organization on the intra- and supramolecular levels, the nature of which is determined by competing interactions of various types. Thus, modification of PNIPAAm by fluorinated comonomers is a promising way of obtaining new copolymers with unique properties.

A fluorine-containing grafted copolymer of polyvinylidenefluoride (PVDF) with PNIPAAm was synthesized [21]. PVDF has many unique properties such as chemical resistance, heat resistance, and good mechanical properties. However, the application of PVDF-based materials is limited due to their hydrophobic nature, while the polymer PNIPAAm, on the contrary, shows hydrophilic properties. PVDF-graft-PNIPAAm copolymers were obtained by the free radical polymerization with transportation of PVDF atom as a macroinitiator and CuCl/tris-(2-(dimethylamino)ethyl) as an amine catalyst. The chemical structure of the grafted copolymers was characterized by IR– and ^1^H NMR spectroscopy. Thermosensitive membranes were obtained from PVDF-graft-PNIPAAm by phase inversion. The influence of temperature on the release of pure water and bovine serum albumin during membrane development was also studied. It was shown that grafted PNIPAAm chains, as a rule, enrich the surface of membranes or membrane pores during their formation. The pore diameter and porosity of the copolymer membranes were greater than those of pure PVDF membranes. In addition, PVDF-PNIPA membranes may exhibit termosensitivity in the analysis of aquatic environments and bovine serum albumin.

Linear-dendritic block copolymers based on *N*-isopropylacrylamide and perfluorinated polyphenylengermane with different MM of hydrophilic blocks were obtained by chain transfer reaction on *bis*-(pentafluorophenyl)germane by radical polymerization of *N*-isopropylacrylamide and subsequent activated polycondensation [22]. The structure of the polymers was confirmed by IR– and ^1^H NMR spectroscopy. The obtained linear-dendritic copolymers were studied in solutions at the water–air interface and in Langmuir-Blodgett (LB) films. It was found that the resulting macromolecules are capable of forming stable monolayer films at the water-air interface, regardless of the subphase acidity. For solutions of PNIPAAm containing a *bis*-pentafluorophenylgermanium group at the end of the chain, no LCST behavior was found. In addition, the introduction of pentafluorophenylgermanium groups into PNIPAAm significantly changes its properties at various interface boundaries.

Previously, using one of the water-soluble monomers of *N*-vinylpyrrolidone, we proved the possibility of synthesis of linear amphiphilic diblock copolymers of poly-*N*-vinylpyrrolidone-2,2,3,3-tetrafluoropropyl methacrylate by the reaction of chain transfer to *bis*-pentafluorophenylgermane during the polymerization of *N*-vinylpyrrolidone and subsequent post-polymerization of selected functional polymers in 2,2,3,3-tetrafluoropropyl methacrylate medium [23]. Properties of copolymers with different MM of hydrophilic block (*M_w_* from 22,000 to 57,000 g/mol) have been studied at various interphase boundaries [24]. It was shown that the behavior of macromolecules of amphiphilic block copolymers in solutions and on the boundary of phases is determined by the formation of hydrogen bonds between the molecules of the solvent and the hydrophobic fluorated fragment. Moreover, fluorine atoms of the perfluorinated phenyl rings participate in the formation of hydrogen bonds at the germanium atom. It was demonstrated that the processing of porous polymer membranes based on different dimethacrylic esters by the solutions of amphiphilic block copolymers of poly-*N*-vynilpyrrolidone-2,2,3,3-tetrafluoropropyl methacrylate in methanol significantly changes their sorption and surface properties [25]. This is what made it possible to test these polymers as modifying components in the production of porous polymer membranes for diesel fuel filtration. It was established that the use of a filter element based on the obtained hydrophobic materials reduces the water content in diesel fuel from 0.54% to 0.27%.

The use of chain transfer reactions during radical polymerization for the synthesis of block copolymers makes it possible to obtain polymers containing blocks of different nature in two stages. The advantage of this method, for example, over RAFT-polymerization, is the ease of synthesis, and the absence of special agents (chain transfer agents).

Interest in amphiphilic copolymers on based PNIPAAm is due not only to their LCST behavior, but also the possibility of the formation of various self-assembly structures in solutions and at interfaces [26,27]. In recent years, the formation of stable Langmuir monolayers of a block copolymer based on PNIPAAm has been studied actively [28,29]. Determining the potential application of new copolymers based on *N*-isopropylacrylamide requires a comprehensive study of properties for the purpose of their directed regulation. This will allow optimizing the choice of polymer materials for solving specific practical problems.

The aim of this work is the study of the behavior of amphiphilic block copolymer of poly-*N*-isopropylacrylamide and poly-2,2,3,3-tetrafluoropropyl methacrylate at the water–air interface (in Langmuir monolayers) and the surface properties of Langmuir-Blodgett films. An important task is the choice of optimal conditions for the formation of monolayers (concentration of the spreading solution and pH of the subphase) and analysis of the stability of multilayer films using surface pressure isotherms obtained under compression-tension conditions.

## 2. Materials and Methods

### 2.1. Materials

The monomer *N*-isopropylacrylamide (Aldrich, 97%, Moscow, Russia) was recrystallized twice from hexane, the crystals were dried in vacuum at room temperature. Then, 2,2,3,3-tetrafluoropropyl methacrylate (FMA) (P&M-Invest, Ltd., 99%, Moscow, Russia) was distilled under reduced pressure (*T* = 48 °C). Methanol-d (Aldrich, 99%, Moscow, Russia) was used without pretreatment. The used solvents (chloroform, methanol, benzene, acetone, hexane, ethanol, and methylene iodide) were purified in accordance with standard procedures [30]. 2,2′-azobisisobutyronitrile recrystallized twice from ethanol, *bis*-(pentafluorophenyl)germane used without pre-cleaning.

### 2.2. Synthesis of Functional Polymers PNIPAAm–Ge(C_6_F_5_)_2_H

To obtain a functional PNIPAAm polymer containing a *bis*-(pentaflourphenyl)germanium group at the end of the chain, a radical polymerization of the NIPAAm was conducted in the presence of 2,2′-azobisisobutyronitrile (AIBN) initiator (0.009 mol/L) in a mixture of solvents and benzene-acetone (1:1) with *bis*-(pentafluorphenyl)germane (Ge(C_6_F_5_)_2_H_2_) ([Ge(C_6_F_5_)_2_H_2_] = 0.02 mol/L, t = 24 h, *T* = 60 °C, [NIPAAm] = 1.3 mol/L) in sealed dilatometer ampules [31] (Figure 1).

The reaction mixtures were degassed in vacuum by triple refreezing, and then soldered and polymerized to 10% conversion. The resulting polymer PNIPAAm–Ge(C_6_F_5_)_2_H was purified from the monomer by tripple reprecipitation using a solvent system: acetone-hexane (1:9 by volume) precipitant, and then dried under reduced pressure and room temperature to constant weight.

### 2.3. Synthesis of Block Copolymers PNIPAAm–Ge(C_6_F_5_)_2_–PFMA

Postpolymerization of PNIPAAm–Ge(C_6_F_5_)_2_H in the FMA medium was carried out up to 12% conversion at *T* = 60 °C, PNIPAAm + FMA = 4% + 96%, [AIBN] = 0.005 mol/L, acetone:FMA = 2:1, [FMA] = 2.08 mol/L [23] (Figure 2). The postpolymerization product was purified by tripple reprecipitation using the solvent system: methanol–hexane precipitator and dried under reduced pressure and room temperature until constant weight.

During the polymerization reaction of functional PNIPAAm–Ge(C_6_F_5_)_2_H in the FMA_,_ not all macromolecules with the active *bis*-(pentafluorophenyl)germanium group participate in the chain transfer reaction. Therefore, the obtained product, which contained, in addition to the PNIPAAm–Ge(C_6_F_5_)_2_H, a block copolymer PNIPAAm–Ge(C_6_F_5_)_2_–PFMA, homopolymers of PNIPAAm and PFMA. To separate them, a hot extraction method was used in the Soxhlet apparatus in specially selected solvents. PFMA was separated from the homopolymer in chloroform, and from PNIPAAm–Ge(C_6_F_5_)_2_H in ethanol.

### 2.4. Polymer Characterization

The IR spectra of the polymers were obtained in KBr tablets on the infrared spectrometer “InfralumFT-801”. Analysis of the structure of the compounds obtained was carried out on ^1^H and ^13^C using the Bruker Avance III 400 NMR spectrometer at *T* = 25 °C (methanol-d was used as the solvent). The spectra were processed using the MestReNova software.

The content of the hydrophilic block in the block copolymer PNIPAAm–Ge(C_6_F_5_)_2_–PFMA was determined using automatic apparatus Kjeldahl Vilitek AKV-20 (Moscow, Russia) designed to measure nitrogen-containing compounds. To accelerate the decomposition reaction, we used a mixed catalyst (10 g HgSO_4_, 7.1 g MgSO_4_ and 0.44 g Se). Polymer samples were decomposed at *T* = 200 °C for 1 h, distillation time 9 min.

### 2.5. Properties of Solution Polymers

The molar masses, hydrodynamic, and conformational characteristics of all samples were determined in solutions in chloroform (density *ρ*_0_ = 1.486 g/cm^3^, dynamic viscosity *η*_0_ = 0.57 cP, and refractive index *n*_0_ = 1.443), methanol (*ρ*_0_ = 0.79 g/cm^3^, *η*_0_ = 0.54 cP, and *n*_0_ = 1.326), chloroform/methanol mixture (9:1) (*ρ*_0_ = 1.460 g/cm^3^, *η*_0_ = 0.56 cP, and *n*_0_ = 1.432) and THF (*ρ*_0_ = 0.890 g/cm^3^, *η*_0_ = 0.46 cP, and *n*_0_ = 1.407) by the methods of light scattering, velocity sedimentation, and GPC triple analysis. All measurements were performed at 21 °C. Weight-average molar masses *M_w_* of polymers investigated were measured by static light scattering using Photocor goniometer (Photocor Instruments Inc., Moscow, Russia). For all samples over the studied concentration range, asymmetry of light scattering intensity was not observed, and the *M_w_* values were found by the Debye method [32] using the formula
(1)$$\frac{cH}{I_{90}}=\frac{1}{M_{w}}+2A_{2}c,$$
where *H* is optical constant:(2)$$H=\frac{4\pi^{2}n_{0}^{2}{\left(dn/dc\right)}^{2}}{N_{A}\lambda^{4}}$$

Here, *c* is the solution concentration, *I*_90_ is the excessive intensity of scattered light at an angle of 90°, *A*_2_ is second virial coefficient, *dn/dc* is the refractive index increment, and *N_A_* is Avogadro’s number. The refractive index increment *dn*/*dc* was measured using a RA–620 refractometer (KEM, Kyoto, Japan). For chloroform solution of PNIPAAm–Ge(C_6_F_5_)_2_H, the second virial coefficient has high positive value; that is, chloroform is a thermodynamically good solvent for this polymer. MM of PFMA was determined in methanol which was a thermodynamically good solvent for this polymer.

The translation diffusion constant *D*_0_ of investigated samples was estimated by dynamic light scattering using the same apparatus as for static light scattering investigation. The correlation function of scattered light intensity was derived with the aid of Photocor-FC correlator with 288 channels (Photocor Instruments Inc., Moscow, Russia). The data were treated by the cumulant method and the Tikhonov regularization procedure. One peak on the intensity distribution has been detected for PNIPAAm–Ge(C_6_F_5_)_2_H in chloroform, PFMA in methanol, and PNIPAAm–Ge(C_6_F_5_)_2_–PFMA in THF. Over the studied concentration range, the hydrodynamic radii *R*_h-D_ of samples macromolecules were independent of concentration *c*. The hydrodynamic radius *R*_h-D_ was calculated using Stoker’s equation:*R*_h-D_ = *kT*/6π *η*_0_D_0_(3)
where *k* is the Boltzmann constant and *T* is the absolute temperature.

A Shimadzu Prominence series GPC system equipped with a refractive index (RI) detector, and a Styragel HR 4E (Waters Associates, Milford, MA, USA) column (7.8 × 300 mm packed with 5 μm particles) was used. The column was calibrated with narrow molecular weight polystyrene standards (purchased from Waters Associates). THF stabilized with 2,6-*tert*-butyl-4-methylphenol (BHT) was used as the mobile phase, at a flow rate of 0.5 mL/min at 40 °C. Three methods for estimation of MM were applied. Firstly, MM were obtained by the refractometric detection (polystyrene standards). Secondly, a combination of refractometric and viscometric detectors was used (universal calibration). Thirdly, a combination of a refractometric and viscometric detector with a light scattering detector (the so-called “triple” detection) was used also.

In a manner similar to [14] velocity sedimentation was studied in chloroform on a MOM-3180 analytical ultracentrifuge (Budapest, Hungary). The rotor rotation speed was 45,000 rpm. The sedimentation boundary was formed artificially by stratifying the solvent on the solution and was recorded with a Philpot–Svensson refractometric optical system. Sedimentation diagrams obtained for all solutions had a unimodal pattern. The sedimentation coefficient *s* was calculated from the rate of movement of the sedimentation boundary. The concentration dependences of *s* are satisfactorily described by the Gralen relationship *s*^−1^ = *s*_0_ −1(1 + *k_s_c*), where *k_s_* is the concentration sedimentation coefficient. Extrapolation of *s*^−1^ to zero concentration made it possible to determine the values of the sedimentation constant *s*_0_ = 6.1 Sv and the parameter *k*_s_ = 42 cm^3^/g.

The density of solvent and solution were measured using a densimeter DA-640 (KEM). The partial specific volume $\bar{v}$ was determined from the slope of the concentration dependences of the difference between the densities of the solution and the solvent. The sedimentation–diffusion molar mass *M*_sD_ was calculated using the Svedberg equation:
(4)$$M_{\mathrm{sD}}=s_{0}RT/(1-\bar{v}\rho_{0})D_{0},$$
where *R* is universal gas constant, *s*_0_ is the sedimentation coefficient at *c* = 0, *ρ*_0_ is density of solvent, *D*_0_ is diffusion constant, *T* is the absolute temperature, and $\bar{v}$ is partial specific volume.

### 2.6. Properties of Monolayers Polymers

Surface pressure isotherms π = f(*A*) were obtained in air using a KSVMini (Finland) apparatus by the Wilhelmy plate method. Solutions of polymers (1 g/L) in a mixture of solvents chloroform/methanol mixture (9:1) were applied to the surface of the subphase with a 50 μL micro syringe in 2 μL portions in a checker board pattern, to evenly distribute the substance over the entire bath surface. The monolayer was held for 30 min to evaporate the solvent from the surface of the subphase, and then compressed at a rate of 10 mm/min. The rate of compression of the monolayers in all experiments was the same and amounted to 10 mm/min. All studies were performed at constant temperature of subphase (21 ± 1) °C. Deionized water with a resistivity of 15 MΩ cm (Spectrum Osmosis water treatment system, Dzerzhinsk, Russia) and 0.1 N HCl solutions were used as a subphase. The area attributable to 1 mg of copolymers in the monolayer (*A*_0_) was determined graphically by extrapolating the descending portion of the isotherm π = f(*A*) on the horizontal axis to π = 0. For the reliability of the result, the surface pressure isotherms for each of the polymers were removed 6 times, and reproducibility was 100%.

The condition in the hysteresis experiment (compression–expansion cycles) was similar to that in the isotherm experiment: after reaching the maximum value surface pressure, the barriers were stopped and monolayer has been expanded with the same speed (the compression rate of the monolayer was 10 mm/min, and the expansion rate was 3 mm/min).

### 2.7. Properties of Langmuir-Blodgett Films Polymers

The monolayer films were transferred using silicon wafers (Telecom-STV JSC, polished, with a specific volume resistance of more than 1 Ω cm, crystallographic orientation 100, Moscow, Russia) as substrates. Before use, the wafers were incubated for 20 min in a chromium mixture, washed further with a flowing, then deionized water and dried at *T* = 80 °C for 1 h. The transfer of monomolecular films from the water–air interface was carried out using the LB technology by KSV minithrough. The transfer conditions were determined from the surface pressure isotherms of the corresponding polymers. For the functional polymer, the films were transferred at a constant surface pressure of 12 mN/m (10 μL), and for the block copolymer at 25 mN/m (30 μL) (*T* = 21 °C and pH = 7). During the deposition, the substrate was dipped into the water (before the spread of the solution) and withdrawn vertically through the monolayer at the speed of 5 mm/min. The transfer coefficient was 0.92–0.98.

Wetting angles θ of LB films were determined under the conditions of leakage by the “sitting drop” method using a setup consisting of a microscope with a light source, a lifting table for a plate and a computer with the “CoolingTech” software. The kinetics of wetting was studied for all films of polymers.

For this purpose, 2 μL of test liquid was applied to the films with a micro syringe, after which, at certain time intervals, we determined the chord of profile *l* and drop height *h*. The measurements were performed until equilibrium was reached (~30 min). The contact angle of wetting θ was calculated in accordance with equation:(5)$$\theta =2\;arctg\;\left(\frac{2h}{l}\right)$$

Gibbs surface energy of LB films $\gamma_{s}$, its polar $\gamma_{s}^{p}$ and dispersion components $\gamma_{s}^{d}$ were determined using the Rukenstein’s approach [33,34,35] (wetting liquids—water and diiodomethane).

The resulting LB films were analyzed by atomic force microscopy (AFM) using Solver P47 microscopes (NT-MDT, Moscow, Russia). The scanning was performed in the tapping mode.

## 3. Results and Discussion

### 3.1. Synthesis and Characterization of Polymers

The relative constant of chain transfer on *bis*-(pentafluorophenyl)germane during radical polymerization of NIPAAm equal to 2.2 determined previously using Mayo method indicates that if it is possible to use this polymer as a basis for production of block copolymers [22]. To obtain an amphiphilic linear block copolymer, a chain transfer reaction was used at the initial stage to obtain a functional polymer PNIPAAm–Ge(C_6_F_5_)_2_H (Figure 1). As a result of its further postpolymerization with 2,2,3,3-tetrafluoropropyl methacrylate (Figure 2), the PNIPAAm–Ge(C_6_F_5_)_2_-PFMA block copolymer was synthesized with a yield of 68.2% and a hydrophilic block fraction of 14.1%. Characteristics of the block copolymer are presented in Table 1.

Figure 1 shows the IR spectrum of the homopolymer PNIPAAm, the functional polymer PNIPAAm–Ge(C_6_F_5_)_2_H and the block copolymer PNIPAAm–Ge(C_6_F_5_)_2_–PPMA. It was shown that the IR spectrum of the functional polymer in comparison with the homopolymer has an absorption band corresponding to the group –CF (1080.5 cm^−1^) and –C_6_F_5_ (960 cm^−1^), which indicates the presence of –GeH(C_6_F_5_)_2_ groups in the macromolecules of the functional polymer. The presence in the IR spectrum of the block copolymer of the absorption bands of –C(O) (1743 cm^−1^) and –CF_2_– (833 cm^−1^) groups, characteristic of PFMA units, indicates the formation of a block copolymer.

Figure 2 shows ^1^H NMR spectrum of a block copolymer. The spectrum of the block copolymer, compared with the functional polymer, indicates the transfer of chemical shifts of the protons of the –CH and –CH_3_ groups to a higher region (the signal 3.96 ppm has shifted to 4.42 and the signal 0.87 ppm shifted to 0.97, respectively). The chemical shifts of the protons of the –O–CH_2_–CF_2_– (4.59 ppm) and –CF_2_H (6.20 ppm) groups also appear in the spectrum (Table 2). The chemical shifts 3.31 and 4.87 (impurity signal) correspond to solvent CD_3_OD. Table 2 also presents the data of ^13^C NMR analysis, according to which the chemical shifts of the carbon atoms of the –CH and –C(O) groups moved to the higher region (44.82 and 175.72 ppm, respectively), and the chemical shift of the carbon atom of the –CH_3_ group to the lower region (18.11 ppm).

The resulting block copolymer PNIPAAm–Ge(C_6_F_5_)_2_–PFMA is insoluble in chloroform; therefore, to study the Langmuir monolayers and to obtain LB films, a mixture of solvents chloroform/methanol was used. However, the presence of methanol in the solvent mixture can have a significant effect on the self-organization of macromolecules at the interphase boundaries, because applying the solution spread on the water–air phase boundary causes chloroform to evaporate, while methanol can either go into the subphase or retain in the loops of macromolecules, as we have previously shown in the case of a block copolymer poly-*N*-vinylpyrrolidone-2,2,3,3-tetrafluoropropyl methacrylate [24].

Molar masses of PNIPAAm–Ge(C_6_F_5_)_2_H determined by static light scattering and velocity sedimentation methods in chloroform and mixed solvent were close (Table 1). Taking into account the high experimental error in determining small values of hydrodynamic radii (up to 20%), it can be assumed that the difference in the *R*_h_ values determined in chloroform and in a mixed solvent is insignificant.

The PFMA sample has a higher molecular weight. Accordingly, a higher value of the hydrodynamic radius was obtained for PFMA. The relatively low value of the refractive index increment for solution PNIPAAm–Ge(C_6_F_5_)_2_–PFMA in chloroform/methanol mixture did not allow reliable determination of MM of this sample by light scattering. Moreover, the compositional heterogeneity of the PNIPAAm–Ge(C_6_F_5_)_2_–PFMA sample should be taken into account. This usually results in an incorrect value of MM at a low value of *dn*/*dc* [36,37,38]. Therefore, MM of block-copolymers was determined by chromatographic method in THF solution. The obtained value of MM is relatively high. It can be assumed that supramolecular structure is formed in THF. This assumption is confirmed by dynamic light scattering data: two modes were detected in solution of PNIPAAm–Ge(C_6_F_5_)_2_–PFMA in THF. Radii of particles responsible for these modes differ ten times. The fast mode reflects the diffusion of macromolecules, while the slow mode reflects aggregate diffusion. As for solutions in a mixed solvent, a high value of *R*_h_ was obtained for them, which may indicate the formation of supramolecular structures.

### 3.2. Langmuir Monolayers of Block Copolymers

The concentrations of spreading solutions were selected and the effect of subphase acidity on the formation of monolayers of macromolecules of the PNIPAAm–Ge(C_6_F_5_)_2_–PFMA block copolymer was studied.

The obtained data show that regardless of the acidity of the subphase, at the studied volumes of the spreading solutions (30–60 μL), high pressure fracture films are characteristic of monolayers collapse pressures (π_max_) = 48–61 mN/m. At small volumes of spreading solutions *V*_s.sol._ = 20 μL, the breakdown pressure of a monolayer film is much higher at the acid subphase (π_max_ = 33 mN/m) compared to water (π_max_ = 13 mN/m).

As an example, Figure 3 shows the block copolymer surface pressure isotherms obtained at *V*_s.sol._ = 30 μL (pH = 7.0 and pH = 1.3). It can be seen that the monolayers undergo several phase transitions, like a three-dimensional gas through “gaseous” (I), “liquid” (II) and “solid” (III) states. When pH = 7.0, these monolayers have anomalic sizes of macromolecules: *A*_0_(pH = 7.0) = 0.6 m^2^/mg, *A*_0_(pH = 1.3) = 0.4 m^2^/mg (Table 3). This could be due to the fact that the acidic subphase triggers the intermolecular interactions between macromolecule–H_2_O.

At large volumes of the spreading solution (for example, *V*_s.sol._ = 50 μL) π_max_ (pH = 1.3) < π_max_ (pH = 7.0): with pH = 7.0 *A*_0_ = 0.3 m^2^/mg, and at pH = 1.3 *A*_0_ = 0.4 m^2^/mg. At the subphase with a pH = 7.0, the breakdown pressure of the monolayer reaches its maximum (π_max_ = 52 mN/m) compared to other *V*_s.sol._; in the case of pH = 1.3 π_max_ = 48 mN/m. Apparently, at such a polymer concentration at the water–air interface, intermolecular interactions begin to predominate over intramolecular interactions, which increases the binding forces between macromolecules in the monolayer and leads to high monolayer collapse pressures.

Apparently, this is due to the ionization of the PNIPAAm blocks, which leads to the unfolding of the hydrophilic block of the macromolecule. At the acidic subphase, intermolecular interactions between macromolecules of the block copolymer predominate over intramolecular interactions, which are confirmed by an increase of the stiffness coefficient of the monolayer β. For example, the value of β for *V*_s_._sol_. = 20 μL at pH = 1.3 is equal to 7.8 × 10^14^ N/m^3^, and in the case of pH = 7.0 it is 5.7 × 10^14^ N/m^3^ (Table 3).

To determine the stability of monolayers, surface pressure isotherms were obtained for the block copolymer in the compression–expansion condition (30 μL). Figure 4 shows that at pH = 7.0, the “pseudoplateaus” region corresponding to the formation of a dense monolayer coincides practically in the compression and expansion cycles. However, this is not observed at pH = 1.3: the π = f(*A*) curve shifts down along the ordinate axis upon stretching. This behavior confirms the presence of significant intermolecular interactions in the case of the acidic subphase. It allowed us to transfer films using the LB technology (*n* = 1) from the subphase at pH = 7.0. The expansion rate of the monolayer was lower than the compression rate.

To adequately analyze the behavior of macromolecules of the amphiphilic block copolymer, the PFMA homopolymer and the functional polymer PNIPAAm–Ge(C_6_F_5_)_2_H (at pH = 1.3 and 7.0) in a chloroform/methanol mixture were stydied (Figure 5). It has been established that when chloroform is used as a solvent, PFMA is characterized by low surface pressure values π_max_ = 14 mN/m and *A*_0_ = 0.2 m^2^/mg (*V*_s.sol._ = 30 μL). In addition, in this case, the surface pressure isotherms obtained under compression conditions manifest the degeneration of the “pseudoplateaus” region, which is the region of formation of a dense monomolecular layer. When chloroform/methanol is used at pH = 7.0 (*V*_s.sol._ = (20–60) μL), all monolayer films of PFMA are characterized by high values of π_max_ = (45 – 60) mN/m and the presence of three phase states of the monolayer (“gaseous” (I), “liquid” (II), “solid” (III) Figure 5). The latter circumstance also indicates the effect of methanol on the behavior of macromolecules at the water–air interface [24].

The surface pressure isotherms of the functional polymer PNIPAAm–Ge(C_6_F_5_)_2_H using a mixture of solvents chloroform/methanol on subphases of different acidity and at different volumes of spreading solutions were obtained also. The curves were compared with similar curves obtained when chloroform was used as a solvent [22]. It was found that in the case of the functional polymer PNIPAAm–Ge(C_6_F_5_)_2_H the presence of methanol in the mixture of the spreading solution does not have a significant effect on the behavior of macromolecules at the water–air interface, that is, only PFMA units participate in the formation of hydrogen bonds.

### 3.3. Surface Properties of LB Films

To determine the Gibbs surface energy of LB films of the block copolymer PNIPAAm–Ge(C_6_F_5_)_2_–PFMA and functional polymer PNIPAAm–Ge(C_6_F_5_)_2_H, the Rukenstein’s approach was used. The surface Gibbs of films ($\gamma_{s}$) and its polar ($\gamma_{s}^{p}$) and dispersion ($\gamma_{s}^{d}$) components were assessed by the Good–Kaelble–Dan–Fowkes equation [33], and water and methylene iodide used as test liquids.

Calculations were performed according to equations
(6){γCH2I2·(1+cosθCH2I2)=2(γCH2I2d·γsd)12+2(γCH2I2p·γsp)12γH2O·(1+cosθH2O)=2(γH2Od·γsd)12+2(γH2Op·γsp)12,
where $\theta_{{\mathrm{CH}}_{2}I_{2}}$ and $\theta_{H_{2}O}$ are the advancing contact angles of methylene iodide and water on the surface of the films. Table 4 shows the results of the wetting of the LB films of the functional polymer and the block copolymer transferred at pH = 7.0.

Table 4 demonstrates that the values of the total Gibbs surface energy of the LB films for the functional and block copolymer are different. In the case of LB film of PNIPAAm–Ge(C_6_F_5_)_2_–PFMA total Gibbs surface energy of the film *γ_s_* = 41.0 mJ/m^2^, which characterizes it as more hydrophobic than the functional polymer film (62.5 mJ/m^2^). Value *γ_s_* of this LB film is close to the total Gibbs surface energy of the LB film of pure PFMA (*γ_s_* = 33.8 mJ/m^2^) [24], that is, in the surface layer of the film predominantly the PFMA blocks remain oriented. For LB film of PNIPAAm–Ge(C_6_F_5_)_2_–PFMA dispersion component (23.9 mJ/m^2^) prevails over the polar component (17.1 mJ/m^2^), and in the case of a functional polymer, on the contrary, the polar component (34.3 mJ/m^2^) is higher than dispersion component (28.2 mJ/m^2^). Therefore, in the case of a block copolymer, dispersion interactions between groups appear in the surface layer of the film –CH(CH_3_)_2_ and –CF_2_CF_2_H, which reduces the wettability of these films with a polar liquid (H_2_O).

The excess of the polar component over the dispersion component in the case of the LB film of functional polymer is most likely due to the predominance of interactions between the water molecules and the polar groups of the –C(O)NH chain over other types of interactions. Practically the same wetting angles of the block copolymer film with water and diiodomethane ($\theta_{H_{2}O\;}$= 64°, $\theta_{{\mathrm{CH}}_{2}I_{2}}$ = 61°) indicate that the nonpolar contact fluid interacts with the surface in almost the same way as the polar one, i.e., in the surface layer of the film, both the hydrophilic units of NIPAAm (–C(O)NH) and the hydrophobic units of PFMA (–CF_2_CF_2_H) are oriented.

The study of the kinetics of wetting showed that for the LB film of PNIPAAm–Ge(C_6_F_5_)_2_H when droplets of test liquids are applied, the change in the wetting angle is Δ$\theta_{H_{2}O}\;$= 6°, and Δ$\theta_{{\mathrm{CH}}_{2}I_{2}}\;$= 1° (30 min). For the LB film of the block copolymer, the change in wetting angles is Δ$\theta_{H_{2}O\;}$= 20° and Δ$\theta_{{\mathrm{CH}}_{2}I_{2}}$ = 1°.

Thus, a comprehensive study of the properties of the obtained polymers based on *N*-isopropylacrylamide at various interphase boundaries showed that the modification of this polymer with a fluorinated linear block is much more efficient than with branched perfluorinated polyphenylenegermane [22], which illustrates the decrease in the surface energy of the LB film (to 41.0 mJ/m^2^) in the case of an amphiphilic block copolymer. Low surface energy of the LB film and the hydrophobicity of the block copolymer PNIPAAm–Ge(C_6_F_5_)_2_–PFMA can be used, for example, to build medical devices with a thromboresistant surface (biosensors, artificial endoprostheses, heart valves and other implants for which the components forming thrombi are unacceptable) [39].

It can be assumed that in the LB film of an amphiphilic block copolymer both links of different nature have contact with the silicon wafer and, based on the above, their orientation in the monolayer can be represented as follows [31].

The AFM images of LB films of functional polymer and block copolymer spread with chloroform/methanol (as an example, transferred at pH = 7) are shown in Figure 6. From Figure 6a, it can be seen that for LB film of functional polymer PNIPAAm–Ge(C_6_F_5_)_2_H exhibit isolated elongated micelles with high densities in the form of “octopus” on the periphery of which there are terminal hydrophobic groups –Ge(C_6_F_5_)_2_H, and the links of the hydrophilic polymer PNIPA are turned inward. When transferring the film LB to a silicon wafer, the entire surface is not covered with a monolayer. For LB film of block copolymer PNIPAAm–Ge(C_6_F_5_)_2_–PFMA (Figure 6b), a comb-like structure is observed with characteristic protrusions that are formed by hydrophobic groups (PFMA and –Ge(C_6_F_5_)_2_H, for example, see Figure 3).

Similar processes of surface micelle formation were discovered by the authors of the work [40] for amphiphilic block copolymers poly(*N*isopropylacrylamide)–*block*–poly[oligo(ethylene glycol) acrylate]. The structure that was first established by the authors for the studied systems differs from the typical core-corona structure consisting of a hydrophobic block core and a hydrophilic block crown in amphiphilic block copolymer systems.

Similar processes of self-organization in Langmuir monolayers and Langmuir-Blodgett films were found for almost all previously obtained amphiphilic block copolymers of various structures [41,42,43,44,45,46,47].

## 4. Conclusions

For the first time, a new amphiphilic block copolymer based on *N*-isopropylacrylamide and 2,2,3,3-tetrafluoropropyl methacrylate was obtained by a double successive chain transfer reaction to *bis*-(pentafluorophenyl)germane groups. The structure of the polymer was confirmed by IR and NMR spectroscopy. The molar masses, hydrodynamic, and conformational characteristics of all polymers were determined in solutions by the methods of light scattering, velocity sedimentation, and GPC triple analysis.

The effect of the acidity of the subphase on the behavior of macromolecules in the Langmuir monolayers of the amphiphilic block copolymer was studied. At pH = 1.3, the formation of monolayers with high surface pressures (π_max_ = 33–61 mN/m) and film stiffness coefficient (β = (4.4–7.8) × 10^14^ N/m^3^) was found, which indicates ionization hydrophilic unit of the block copolymer macromolecule on the acidic subphase. The influence of methanol on the self-organization of fluorinated units at the water–air interface, which is capable of forming hydrogen bonds with macromolecules, has been revealed. Results provide insight into the two-dimensional self-assembly of amphiphilic block copolymers consisting of molecular-level amphiphilic block components.

## Data Availability

Not applicable.

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
