# Peer review of "Properties in Langmuir Monolayers and Langmuir-Blodgett Films of a Block Copolymer Based on N-Isopropylacrylamide and 2,2,3,3-Tetrafluropropyl Methacrylate"

_polymers, 2022, doi:10.3390/polym14235193_

Round 1

Reviewer 1 Report

the ms in my opinion is borderline. There are so many typos and important experiments are needed as highlighted in the pdf herein attached.

Author Response

Dear reviewer. Thank you very much for your careful work with our article.

 Responses to the comments of the Reviewer 1 (remarks were in the file pdf).1. Not belonging to Intro the introduction (9 first lines).            P. 1. We agree with the comment, the introduction has been corrected.2. Define all terms of equation

We corrected text (we added “where R is universal gas constant, s0 is the sedimentation coefficient at c = 0, r0 is density of solvent, D0 is diffusion constant, T is the absolute temperature, and  is partial specific volume”).

   P. 6  We agree with comment. The formula was corrected.3. Why the choice of 12 and 25 mN/m? P. 6. The transfer conditions were determined from the surface pressure isotherms of the corresponding polymers. For the functional polymer, the films were transferred at a constant surface pressure of 12 mN/m (10 μl), for the block copolymer at 25 mN/m (30 μl) (T = 21 °C and pH = 7).

This comment has been added to the text of the manuscript on p. 6.

  1. Figure should appear after related text and not before
  2. 8, P. 9, and P.10. We agree with comments. The text describing the pictures has been moved before the pictures.
  3. I wonder whether this section about synthesis could moved to a supporting info section. Figure 3 should come before Table 3 not after
  4. 10. In principle, we agree with the remark. But we decided that for a better understanding by the reader, leave this information in this section. Since there is no more data for Supporting information. The table was moved.
  5. The authors should explain also the effect of the barrier speed to the Langmuir monolayer formation Explain the Rukenstein approach and provide related reference! Effect of pH on LB monolayer formation?
  1. P. 11. In this work, all experiments on monolayer compression were carried out under the same conditions at a compression rate of 10 mm/min. Added on the page 6 (сhapter 2.6) to the corresponding section of the experimental part.
  2. P. 13. We agree with the remark, the system of Equations on which the calculations were carried out was added to the text (P. 14).
  3. P. 14. The effect of pH on the formation of LB films is not discussed in this work, since a separate article will be devoted to this in the near future. Here, as an example, a film LB transferred at pH = 7 is given.

Reviewer 2 Report

The following article entitled ‘Properties in Langmuir monolayers and Langmuir-Blodgett films of a block copolymer based on N-Isopropylacrylamide and 2,2,3,3-tetrafluropropyl methacrylate’ authored by Zamyshlyayeva et al is a novel and nice work. In my opinion it is suitable to be published in Polymers while the first part of the introduction should be removed before that. The first 9 lines appear redundant and not at all part of introduction.

Author Response

Responses to the comments of the Reviewer 2. 

Dear reviewer. Thank you very much for your careful work with our article.

  1. The following article entitled ‘Properties in Langmuir monolayers and Langmuir-Blodgett films of a block copolymer based on N-Isopropylacrylamide and 2,2,3,3-tetrafluropropyl methacrylate’ authored by Zamyshlyayeva et al is a novel and nice work. In my opinion it is suitable to be published in Polymers while the first part of the introduction should be removed before that. The first 9 lines appear redundant and not at all part of introduction.

            We agree with the remark. The first 9 lines from the introduction were removed.

Reviewer 3 Report

The manuscript "Properties in Langmuir monolayers and Langmuir-Blodgett films of a block copolymer based on N-Isopropylacrylamide and 2,2,3,3-tetrafluoropropyl methacrylate" by Olga Zamyshlyayeva et al is of potential interest. Indeed, the study of the interfacial properties of newely synthetized amphiphilic block copolymer may be interesting for medical applications.

However, I found critical points that should be addressed to the Authors.

1.     The first nine lines of the “Introduction” should be deleted. They probably come from an article template. A similar situation occurs at the end of this article with the Author Contributions section. The first paragraph should be deleted as it does not apply to the Authors of the reviewed manuscript.

2.     The determination of the collapse point of the monolayer is questionable for me. The collapse point is usually recognized as the point at which the first derivative of the isotherm begins to decrease, or as the point at which the isotherm begins to deviate from a steep slope, representing the condensed phase of monolayer. In the case of the presented research, the monolayer collapse corresponds to the last point of the isotherm, where we can actually expect a multilayer system.

3.     Page 11, 1st paragraph, author stated: It can be seen that the monolayers undergo several phase transitions, like a three-dimensional gas through “gaseous”, “liquid” and “liquid-crystalline states”. All these phase transitions should be clearly identified on the isotherms.

4.     What was the reproducibility of the isotherms?

5.     Figure 4: in order to accurately compare the hysteresis of isotherms, both films had to be compressed to the same value of the surface pressure, i.e. 45 mN/m. Moreover, the expansion speed should be lower than the compression speed.

6.     Figure 3-5: replace commas with dots on the x-axis scale.

Summing up, I would like to say that I highly appreciate the chapter devoted to the synthesis and characterization of the studied polymers. These results may be useful to other researchers. However, my main problem in this work is the limited reliability of information obtained from surface pressure isotherms.

Author Response

Responses to the comments of the Reviewer 3.

Dear reviewer. Thank you very much for your careful work with our article.

 1. The first nine lines of the “Introduction” should be deleted. They probably come from an article template. A similar situation occurs at the end of this article with the Author Contributions section. The first paragraph should be deleted as it does not apply to the Authors of the reviewed manuscriptWe agree with the remark. Sections Introduction and Authors Contributions have been modified.2. The determination of the collapse point of the monolayer is questionable for me. The collapse point is usually recognized as the point at which the first derivative of the isotherm begins to decrease, or as the point at which the isotherm begins to deviate from a steep slope, representing the condensed phase of monolayer. In the case of the presented research, the monolayer collapse corresponds to the last point of the isotherm, where we can actually expect a multilayer system.

We agree with the remark that not all the curves presented in the work clearly show the point of collapse, therefore, throughout the text of the article, the fracture pressure (πcol) was replaced by the maximum pressure (πmax).

  1. Page 11, 1stparagraph, author stated: It can be seen that the monolayers undergo several phase transitions, like a three-dimensional gas through “gaseous”, “liquid” and “liquid-crystalline states”. All these phase transitions should be clearly identified on the isotherms.

As for the phase transitions in the Figures. Changes have been made to Figures 3 and 5, in which these transitions are marked (I - gaseous, II - liquid, III - solid). Relevant comments are included in the text of the article (on p. 12 and 13).

  1. What was the reproducibility of the isotherms?

For the reliability of the result, the surface pressure isotherms for each of the polymers were removed 6 times, reproducibility was 100%. (P. 6).

  1. Figure 4: in order to accurately compare the hysteresis of isotherms, both films had to be compressed to the same value of the surface pressure, i.e. 45 mN/m. Moreover, the expansion speed should be lower than the compression speed.

Figure 4 shows isotherms under compression-expansion conditions. The conditions were determined from the respective surface pressure isotherms for polymers obtained under compression conditions (for pH 7.0 – 45 mN/m, for pH 1.3 – 50 mN/m). The speed of compression and stretching was different. This was noted in the experimental part (p. 6).

  1. Figure 3-5: replace commas with dots on the x-axis scale.

We agree with the comments, the commas in Figures 3-5 have been replaced with dots.

Reviewer 4 Report

The authors synthesized a copolymer of poly(Nisopropylacrylamide)Ge(C6F5)2poly(2,2,3,3-tetrafluoropropyl methacrylate) by using two radical polymerization bridged by GeH2(C6H5)-mediated chain transfer reaction. The resultant amphiphilic copolymer was employed to prepare films and their properties were well studied. Although the method and result are interesting, some necessary revisions are required before being published.

1.     A chain transfer reaction was employed to construct the deblock copolymer. The authors are suggested to give a brief introduce on chain transfer, especially the advantage features to the other coupling reaction.

2.     The chain transfer process is not clear, for example, efficiencies of the capping reaction and re-initiation process were not given, and thus it is difficult to confirm the product is a copolymer or a mixture of two homopolymers. The process should be monitored by using NMR and GPC, especially, GPC curves of product after first and second radical polymerization, as well as purification should be given to confirm all the two homopolymers of PNIPAAm and PFMA were removed.  

3.     Molecular weight plays an important role in self-assembly. The PDIs of the PNIPAAm and PNIPAAm-PFMA should be given and their influence on morphology should be discussed.

4.     There some typos should be checked,

1)     the start of introduction should be checked;

2)     “spectra” in the capping of Figure 2 should be “spectrum”;

3)     typesetting should be doble checked.

Author Response

Responses to the comments of the Reviewer 4.

Dear reviewer. Thank you very much for your careful work with our article.

  1. A chain transfer reaction was employed to construct the deblock copolymer. The authors are suggested to give a brief introduce on chain transfer, especially the advantage features to the other coupling reaction.

In the text of the article, they added about the advantages of the approach used (P. 3).

  1. The chain transfer process is not clear, for example, efficiencies of the capping reaction and re-initiation process were not given, and thus it is difficult to confirm the product is a copolymer or a mixture of two homopolymers. The process should be monitored by using NMR and GPC, especially, GPC curves of product after first and second radical polymerization, as well as purification should be given to confirm all the two homopolymers of PNIPAAm and PFMA were removed.  

Yes, indeed, during the reaction, the formation of homopolymers is possible. But the resulting reaction products were characterized by NMR spectroscopy. In addition, the cured polymers were carefully separated by hot extraction (in the text on p. 4).

  1. Molecular weight plays an important role in self-assembly. The PDIs of the PNIPAAm and PNIPAAm-PFMA should be given and their influence on morphology should be discussed. There some typos should be checked. The start of introduction should be checked; “spectra” in the capping of Figure 2 should be “spectrum”; typesetting should be double checked

We agree with the remark that molecular weight plays an important role in the self-organization of macromolecules.The PDIs for PNIPAAm-PFMA is 2.3.  It is necessary to carry out special studies on the synthesis of samples that differ greatly in terms of degree of polydispersity, but are identical in composition, which certainly is, and at the same time have similar MM values. We hope to carry out this work in the future. Based on the results of studies of single samples, it is impossible to draw such a conclusion. These typos in the text have been corrected.

As for the references, we added 7 references to decrease our percentage of citations in the paper.

All changes which were made to the article are highlighted in yellow.

Round 2

Reviewer 1 Report

I'm happy with the authors' answers and with the revised ms. I think the ms is now acceptable for publication.

Reviewer 4 Report

The manuscript was improved.